# BioTemporal-HAR: Optimizing Behavioral Biometrics through Signal Detection Theory and Multi-Sensor Fusion

Louis Rollet

Vincent Montero Fontaine

Time Series Intelligence, Spring 2026

April 28, 2026

## Abstract

Human Activity Recognition (HAR) has moved beyond coarse activity labeling toward behavioral biometrics, where the temporal structure of motion can support recognition of both what a user is doing and who the user may be. This proposal introduces BioTemporal-HAR, a lightweight multi-sensor framework for recognizing activities and biometric identity from smartphone and smartwatch inertial signals. The project combines windowed signal processing, feature-selection experiments, a compact CNN-LSTM sequence model, and Signal Detection Theory (SDT) calibration to improve accuracy, reduce computational cost, and explicitly evaluate false alarm behavior. We target the WISDM Smartphone and Smartwatch Activity and Biometrics dataset, which contains accelerometer and gyroscope streams for 51 subjects performing 18 activities. The expected outcome is a reproducible pipeline that evaluates activity recognition, closed-set identity classification, and identity verification under edge-deployment constraints.

## 1 Introduction

Human Activity Recognition (HAR) is a time-series classification problem in which wearable or mobile sensors capture the dynamics of human movement. Early HAR systems focused on broad activity labels such as walking, sitting, and standing. Newer behavioral biometrics systems use the same inertial signals as continuous, passive identity cues: the rhythm of a gait cycle, the wrist motion in typing, or the acceleration pattern of eating can act as a personal motion signature (WISDM Lab 2019).

BioTemporal-HAR treats smartphone and smartwatch signals as both activity indicators and biometric evidence. The project will classify 18 daily activities and evaluate two biometric settings: closed-set identification among the 51 known subjects and identity verification, where a claimed identity is accepted or rejected using a calibrated score. This dual objective makes the problem more realistic than closed-set activity recognition alone because it must model activity-dependent variation, person-specific motion, and uncertainty in the final decision.

The central research question is whether a lightweight multi-sensor model can preserve biometric detail while remaining efficient enough for wearable deployment. We therefore combine compact temporal modeling with Signal Detection Theory (SDT), which provides explicit tools for separating discriminability from decision bias (Green and Swets 1966).

## 2  Motivation

Wearable devices enable passive authentication and health monitoring without requiring users to interrupt their daily routines. Traditional biometric systems such as fingerprints or facial recognition require active presentation and may be inconvenient in clinical, workplace, or assistive settings. Behavioral biometrics instead operate continuously by observing natural motion patterns (WISDM Lab 2019).

The problem also has safety-critical implications beyond the specific WISDM labels. In elder care or rehabilitation, distinguishing ordinary walking from instability would be not only a classification problem but also a detection problem; however, WISDM does not directly contain fall or pre-fall labels, so this proposal treats that scenario as motivation rather than as a claim the dataset can validate. Most HAR pipelines report accuracy or F1-score, but those metrics do not specify how the model should trade off misses against false alarms. SDT gives the project a principled way to tune the decision criterion for security-sensitive or health-sensitive contexts.

## 3  Background and Related Work

HAR systems usually segment raw accelerometer and gyroscope streams into overlapping windows, extract time-domain or frequency-domain features, and train a classifier on those windows. Classical methods such as support vector machines and random forests can perform well with handcrafted statistics, but they may be less suited to longer temporal dependencies and user-specific structure without careful feature design (Anguita et al. 2013).

Deep learning models reduce the need for manual feature engineering. Convolutional layers capture local temporal motifs, while recurrent layers such as LSTMs model order and persistence in sequential signals (Hochreiter and Schmidhuber 1997). Hybrid CNN-LSTM models have become a strong baseline for wearable HAR because they combine local filtering with temporal memory (Ordonez and Roggen 2016). However, depending on their depth and hidden-state size, these networks can become too large for practical deployment on low-power devices.

Feature selection and model compression are therefore important for this project. Meta-heuristic search methods such as genetic algorithms and particle swarm optimization have been used to reduce feature spaces and tune model configurations. We propose to investigate the Walrus Optimizer as a candidate feature-selection strategy for selecting informative time-frequency features, while comparing it with simpler baselines such as mutual information, random forest importance, or recursive feature elimination (ResearchGate 2025).

## 4  Problem Definition

The project addresses three coupled tasks.

First, the system must classify activities from inertial time series. Given a window of synchronized sensor readings, the model predicts one of 18 activity labels. Second, the system must identify the subject associated with the motion pattern in a closed-set setting. Third, the system must calibrate verification decisions so that a target false alarm rate can be evaluated when the model decides between a genuine claimed identity and an impostor claim.

Several challenges make the task non-trivial. Phone and watch signals have different noise profiles because the devices are worn at different body locations. We hypothesize that

fine-grained actions such as eating or typing will be more visible at the wrist, whereas gait patterns may be clearer from the pocket, but this assumption will be tested through sensor ablations. The dataset is also high-dimensional and may be imbalanced after windowing, so class frequency and per-subject coverage will be reported before training.

## 5   Proposed Method

The BioTemporal-HAR pipeline has four stages: preprocessing, feature selection, sequential modeling, and SDT calibration.

### 5.1   Preprocessing and Windowing

Raw tri-axial accelerometer and gyroscope streams will be cleaned, synchronized by device and subject, and segmented into fixed-length windows. The WISDM raw streams are sampled at 20 Hz, so a 2.56 second window contains about 51 samples rather than the 128 samples used by 50 Hz HAR datasets such as UCI HAR (Anguita et al. 2013). We will therefore report each window length in both seconds and samples, and we will test window sizes such as 2.5 seconds and 5-10 seconds with 50 percent overlap. A low-pass Butterworth filter will be considered when validation results show that separating low-frequency body motion from high-frequency noise is beneficial.

### 5.2   Feature Selection

The project will compute descriptive time-domain and frequency-domain features for each window, including mean, variance, energy, entropy, correlation, and spectral coefficients. Feature selection will be used primarily for classical feature-based baselines and for evaluating whether compact feature sets preserve activity and identity information. If selected features are fed into a neural model, the architecture will be adjusted so that parameter-count reductions are measured directly rather than assumed from feature reduction alone.

### 5.3   CNN-LSTM Architecture

The model will use one-dimensional convolutional layers to extract local temporal patterns from each sensor stream, followed by an LSTM layer to model longer dependencies. Residual connections will be considered to stabilize training and reduce the need for deeper recurrent stacks. The architecture will be intentionally small so that it can act as a realistic edge-device baseline rather than only a high-capacity research model.

### 5.4   Multi-Sensor Fusion

Phone and watch representations will be fused after stream-specific feature extraction. A lightweight attention or gating mechanism will estimate which device contributes the strongest evidence for each activity window. We expect wrist motion to help with some fine-grained hand activities and phone motion to help with locomotion and gait-based cues, but this claim will be evaluated through phone-only, watch-only, and fused-model ablations.

### 5.5   SDT Calibration

The final identity module will be evaluated as a detection system. In verification experiments, genuine claimed-subject windows will be treated as signal trials and impostor claims as noise trials. For each identity score, SDT metrics will estimate hit rate, false alarm rate, miss rate, sensitivity ($d'$), and criterion ($c$). Decision thresholds will be selected on a validation split

to target low false alarm behavior, then tested on held-out data. This separates the model's ability to discriminate users from the policy used to accept or reject a claimed identity.

# 6 Dataset

The project will use the WISDM Smartphone and Smartwatch Activity and Biometrics dataset (WISDM Lab 2019). It contains inertial data from 51 subjects performing 18 activities, including walking, jogging, typing, eating, and other daily movements. Both smartphones and smartwatches provide accelerometer and gyroscope signals, creating a natural setting for cross-device fusion.

The scale of the dataset, with more than 15 million raw sensor readings, is large enough to support deep learning experiments while remaining manageable on a standard GPU or Google Colab environment.

# 7 Evaluation Plan

We will evaluate the system with activity recognition metrics, biometric identification metrics, and detection metrics. Activity performance will be measured with accuracy, macro F1-score, and confusion matrices to expose class imbalance. Identity performance will be measured with top-1 accuracy and per-subject error rates. Detection behavior will be measured with ROC-AUC, false alarm rate, miss rate, and SDT-derived estimates of $d'$ and $c$.

To reduce leakage from overlapping windows, train, validation, and test partitions will be created before window generation whenever possible, using grouped splits by subject files, activity trials, or time ranges. For activity recognition, we will report whether the evaluation uses subject-overlapping or subject-held-out splits. For closed-set identity classification, the test set will contain held-out windows from enrolled subjects. For verification, the test set will contain both genuine claims and impostor claims drawn from held-out windows.

The main experimental comparisons will be:

1. classical feature-based baselines versus CNN-LSTM models;
2. phone-only, watch-only, and fused sensor configurations;
3. full feature sets versus feature subsets selected by Walrus and simpler baseline methods;
4. default softmax decisions versus SDT-calibrated thresholds.

# 8 Research Targets

The project will test four measurable targets rather than assume they are achievable before experimentation.

1. Evaluate whether a lightweight CNN-LSTM can approach or exceed 95 percent activity accuracy under leakage-safe splits.
2. Evaluate whether movement patterns can support more than 90 percent top-1 closed-set identity accuracy on held-out windows from enrolled subjects.
3. Test whether identity verification can reach a false alarm rate below 1 percent while reporting the corresponding miss rate and threshold criterion.
4. Measure whether feature selection, channel selection, or compact architecture choices can reduce model parameters or input dimensionality by approximately 50 percent compared with a larger BiLSTM baseline.

## 9 Project Schedule

| Week | Dates | Main tasks | Responsibility |
|------|-------|-----------|----------------|
| 1 | Apr 22-28 | Review SDT and metaheuristic feature selection; clean and inspect WISDM data. | Louis and Vincent |
| 2 | Apr 29-May 5 | Implement 1D-CNN, LSTM, and ResLSTM baselines. | Shared development |
| 3 | May 6-12 | Run feature-selection experiments and add SDT threshold calibration. | Shared development |
| 4 | May 13-19 | Complete sensor-fusion ablations, analyze results, and prepare the final report. | Shared writing and analysis |

## 10 Current Status

The dataset has been acquired and its integrity has been checked. Initial exploratory analysis suggests that wrist-based sensors may be more informative for hand-centered activities such as typing and eating, while pocket-based sensors may better capture gait-related patterns; these observations will be verified with formal ablation experiments. The current implementation work is focused on setting up the Google Colab environment and building the first 1D-CNN feature extractor.

## 11 Risks and Limitations

Several limitations must be considered when interpreting the results. WISDM is a controlled dataset rather than a fully naturalistic deployment study, so performance may not transfer directly to unconstrained real-world use. The dataset supports activity recognition and behavioral biometrics, but it does not directly validate fall detection, pre-fall detection, or clinical decision-making. Identity results may also be sensitive to device placement, subject demographics, repeated trials from the same collection protocol, and train-test leakage from overlapping windows. Finally, behavioral biometric systems raise privacy and consent concerns because they can infer identity from passive motion traces, so any deployment claim should be framed cautiously.

## 12 Conclusion

BioTemporal-HAR proposes a compact and calibrated approach to wearable behavioral biometrics. By combining multi-sensor fusion, feature selection, sequential modeling, and SDT-based thresholding, the project aims to produce a model that is not only accurate but

also interpretable in terms of detection risk. The result should be a reproducible HAR pipeline for activity recognition and passive identity verification, with clear limits on what the WISDM dataset can and cannot support.

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
