# OpenReview forum: "BioTemporal-HAR: Optimizing Behavioral Biometrics through Signal Detection Theory and Multi-Sensor Fusion"
_tsinghua.edu.cn/THU/2026/Spring/ANM — THU 2026 Spring ANM Submission_

### Official Review · Reviewer_gbCD · 2026-05-14

**Rating:** 9
**Confidence:** 4

**Summary:**

The paper proposes BioTemporal-HAR: multi-sensor framework for recognizing activities from biometric inertial signals, including smartwatch and smartphone signals. Expected outcomes include a reproducible pipeline that improves on existing HAR pipelines by specifying how the model can handle false alarms. Inertial data from 51 subjects performing 18 different movement activities in dataset "WISDM Smartphone and Smartwatch Activity and Biometrics dataset" (WISDM Lab 2019) is utilized.

**Strengths:**

Strong introduction to the background of Human Activity Recognition, providing enough detail in a concise enough manner so as to not take up too much content while maintaining informational context. Allows reader to understand the time series problem of HAR, a recognition problem relating to wearable device sensors for human movement.

Very clear research targets with quantifiable metrics. Interesting problem target to existing papers regarding the false-alarm model response research.

**Weaknesses:**

Dataset is from 2019, which is 7 years ago.

Suggestion (grammar): in Section 1 Evaluation Plan:
Capitalize the starting line of each of the 4 points under "main experimental comparisons".

**Questions:**

Asking this in a curious rather than critical way, but what led to the choice of using measuring metrics such as:
- Macro F1-score,
- Confusion matrices,
To expose class imbalance?

---

### Official Review · Reviewer_PFAe · 2026-05-14

**Rating:** 7
**Confidence:** 5

**Summary:**

The authors propose BioTemporal-HAR, a lightweight multi-sensor framework for Human Activity Recognition and behavioral biometrics using smartphone and smartwatch inertial signals. The project uses the WISDM Smartphone and Smartwatch Activity and Biometrics dataset, aiming to classify 18 activities, perform closed-set subject identification, and conduct identity verification using Signal Detection Theory (SDT). The proposed pipeline includes preprocessing/windowing, handcrafted time-frequency features, feature selection, a compact CNN-LSTM model, phone-watch sensor fusion, and SDT-based threshold calibration for verification.

**Strengths:**

- Clear Motivation: The proposal is well-motivated. It connects HAR with behavioral biometrics and explains why passive motion signals can be useful for both activity recognition and identity verification.
- Strong Evaluation Awareness: The authors correctly identify train-test leakage from overlapping windows as a major risk and propose grouped splits before window generation when possible. This is important for time series evaluation.
- Good Use of SDT: The use of SDT is a strong part of the proposal. Instead of only reporting accuracy or F1-score, the authors plan to evaluate false alarms, misses, sensitivity, and decision criterion, which is more appropriate for verification and security-related applications.
- Comprehensive Ablation Plan: The planned comparisons are meaningful, including phone-only vs. watch-only vs. fused models, classical feature baselines vs. CNN-LSTM, full features vs. selected features, and default softmax decisions vs. SDT-calibrated thresholds.

**Weaknesses:**

- Missing Architecture Visualization: The proposal is clear in text, but there is no system diagram. A diagram showing the phone/watch streams, feature extraction, CNN-LSTM branches, fusion module, and SDT calibration would make the method much easier to understand.
- Fusion Mechanism Is Still Vague: The authors mention a lightweight attention or gating mechanism, but the exact formulation is not defined. It is unclear whether fusion happens at the feature level, hidden-state level, or score level, and how the model decides which sensor is more useful for each activity.

**Questions:**

- How exactly will the CNN-LSTM be trained for the three tasks: one shared encoder with multiple heads, or separate models for activity, identity, and verification?
- What is the exact formula or architecture for the phone-watch fusion mechanism?

---

### Official Review · Reviewer_fUt3 · 2026-05-15

**Rating:** 8
**Confidence:** 4

**Summary:**

This proposal introduces a lightweight multisensor framework to recognize 18 different activities in 51 subjects both in a closed-set identification and identity verification settings. The dataset was collected in 2019 from sensors in smartphone and smartwatch. The core working principle is CNN-LSTM in combination with SDT calibration.

**Strengths:**

- integration with SDT is better in sensitive scenarios than simple metric evaluation like F1 score
- lightweight model for edge devices
- rigorous evaluation plan

**Weaknesses:**

- main problem is with the dataset. If the goal is to detect fall detection or some sensitive clinical scenarios, this dataset cannot utilize the full capacity of your approach.
maybe you can look into dataset like this one?
https://github.com/tashik19/SmartFallMM-Dataset/

**Questions:**

What is the reason that you specfically choose this dataset?

---

### Official Review · Reviewer_o2Cd · 2026-05-16

**Rating:** 7
**Confidence:** 5

**Summary:**

Bio Temporal-HAR proposes a lightweight multi-sensor framework for Human Activity Recognition (HAR )and behavioral biometrics using smartphone/smartwatch inertial data from the WISDM dataset. It combines windowed signal processing, Walrus Optimizer-based feature selection, a compact CNN-LSTM, sensor fusion, and SDT-based calibration for activity classification, identity recognition, and verification, targeting edge deployment.

**Strengths:**

The proposal integrates feature selection, CNN-LSTM modeling, multi-sensor fusion, and SDT calibration into a practical, edge-ready pipeline with thorough evaluation metrics and sensor ablation analysis. Clear, measurable targets (95% activity accuracy, ~50%parameter reduction) add ambition and verifiability.

**Weaknesses:**

The references used in this proposal are very few, which indicates the literature review is not good enough. I'm sure WISDM Lab 2019) and(ResearchGate 2025) are not author names for those paper you are citing.The controlled WISDM dataset limits real world generalizability. Privacy implications of behavioral biometrics are noted but under explored.

**Questions:**

1.  How will you ensure generalizability beyond the WISDM dataset, given its controlled collection environment?

---

### Official Review · Reviewer_gyK6 · 2026-05-16

**Rating:** 8
**Confidence:** 4

**Summary:**

This proposal introduces BioTemporal-HAR, a lightweight multi-sensor framework for recognizing activities and evaluating biometric identity from smartphone and smartwatch time series. The overall pipeline includes windowed signal processing, feature-selection experiments, CNN-LSTM architecture, multi-sensor fusion, and signal detection theory. The dataset (Biometrics, 2019) contains accelerometer and gyroscope signals from 51 individuals and classify 18 different activities.

**Strengths:**

- Original topic which background and real-life applications have been clearly exposed.
- Solid framework pipeline. The multi-sensor fusion and the SDT parts are especially well explained.
- Relevant experimental comparisons, which should be enough for the authors to identify the strengths and weaknesses of their model.
- Risks and limitations have already been identified, thus refraining the authors from drawing hasty conclusions.

**Weaknesses:**

- The reference part could have been more expanded, especially by quoting more HAR state-of-art works.
- The choice of the evaluation metrics could have been more detailed.

**Questions:**

Apart from CNN-LSTM, which other architectures could have been used for the sake of HAR? Why did you decide to go with CNN-LSTM at the end?

---

### Official Review · Reviewer_vAzQ · 2026-05-17

**Rating:** 8
**Confidence:** 4

**Summary:**

This proposal introduces BioTemporal-HAR, a lightweight multi-sensor framework designed to handle both human activity recognition (HAR) and behavioral biometrics using inertial signals from smartphones and smartwatches. The methodology combines windowed signal processing, feature selection (including the Walrus Optimizer), a compact CNN-LSTM architecture, and Signal Detection Theory (SDT) calibration. Utilizing the WISDM dataset, which includes data from 51 subjects performing 18 activities, the project aims to evaluate activity recognition, closed-set identity classification, and identity verification. The authors set explicit, measurable targets, such as achieving >95% activity accuracy and <1% false alarm rates for verification, all while maintaining edge-deployment efficiency.

**Strengths:**

Clear and Practical Motivation: The proposal excellently justifies the shift from coarse HAR to continuous behavioral biometrics, noting its value in passive authentication and health monitoring without interrupting daily routines.

Innovative Use of SDT: Integrating Signal Detection Theory (SDT) is a major strength. Instead of relying solely on standard accuracy or F1-scores, SDT provides a principled method to tune decision thresholds and balance false alarms against misses, which is critical for security and health contexts.

Rigorous Evaluation Design: The experimental plan is exceptionally well-structured, clearly delineating between activity recognition, closed-set identification, and verification, while also taking precautions to avoid train-test leakage from overlapping windows.

Concrete Research Targets: Unlike many proposals that promise general improvements, this document outlines four highly specific, measurable benchmarks for success (e.g., specific accuracy thresholds and a 50% parameter reduction target).

**Weaknesses:**

While the proposal mentions using a "lightweight attention or gating mechanism" to fuse watch and phone data, it lacks technical detail on how this will be implemented without undermining the goal of keeping the model small.

The proposal introduces the Walrus Optimizer for feature selection, but does not provide strong justification for why this specific, novel algorithm is hypothesized to outperform the mentioned simpler baselines (mutual information, random forests) in this specific domain.

**Questions:**

1. How exactly will the lightweight attention or gating mechanism be structured to ensure the CNN-LSTM model remains small enough for practical edge deployment?

2. Given that the proposal anticipates potential class imbalance after windowing, what specific techniques (e.g., loss weighting, resampling) will be used during the training phase to mitigate this?

---

### Official Review · Reviewer_ChMG · 2026-05-17

**Rating:** 8
**Confidence:** 4

**Summary:**

BioTemporal-HAR is introduced as a lightweight multi-sensor model for recognizing daily activities and biometric identity based on inertial signals. The framework involves windowed signal pre-processing, a compact CNN-LSTM sequence model, and Signal Detection Theory (SDT) calibration. The dataset is drawn from 51 subjects performing 18 daily actions recorded on both accelerometers and gyroscopes in the  subjects' smartphones and smart-watches, enabling cross-device fusion.

**Strengths:**

- The problem is well-motivated and its importance is conveyed clearly
- The pipeline is generally well-formulated and justified by prior research
- It's a good idea to separately evaluate based on SDT
- The evaluation plan and research goals are laid out clearly

**Weaknesses:**

- The Walrus optimizer was introduced without explanation or clear justification
- The related works section could be more developed (could have introduced BiLSTM application and limitations for HAR before later using it as benchmark for model size)

**Questions:**

- Will impostor trials draw from only held-out subjects, or will enrolled subjects act as impostors for other enrolled identities? How does this choice affect the real-world interpretation of your false alarm rate?

---

### Official Review · Reviewer_Sfjr · 2026-05-18

**Rating:** 8
**Confidence:** 4

**Summary:**

The submission proposes BioTemporal-HAR framework for performing behavioral biometric identification by leveraging signals from smartphones and smartwatches across 18 activities documented in the WISDM dataset. The work combines compact CNN-LSTM modeling with Signal Detection Theory (SDT) calibration for accomplishing three tasks: activity classification, closed-set identification, and identity verification.

**Strengths:**

- CNN-LSTM architecture is designed with real-world deployability in mind, considering both broader computational constraints and domain-specific application (tracing both individual-specific and movement type-specific variations);
- Evaluation framework and research targets are concrete and measured against existing benchmarks;
- A very clear formulation of the project around time series classification.

**Weaknesses:**

- A careful review of the citations and related work could strengthen the contextual foundation of the project;
- The core dataset seems to consist of isolated activities of individuals, while real-world applications may possess a more varied sensor noise, include sequential or overlapping activities co-influencing each other;
- Some of the methodological choices have a limited grounding on their selection for the given problem (i.e. the Walrus optimizer).

**Questions:**

- The WISDM dataset largely contains isolated activities; how do you anticipate the system might perform on sequences of multiple or overlapping activities? Additionally, how robust is the recognition in the presence of sensor noise or variability inherent to real-world deployments?
- Your proposed approach focuses on detecting and learning individual motion patterns, which could be relevant for healthcare anomaly detection, as you outline in the motivation section. Do you plan to explore extensions of this framework for clinical or health-focused applications, i.e. fall detection or behavioral anomalies?

---

### Official Review · Reviewer_UPjg · 2026-05-18

**Rating:** 4
**Confidence:** 4

**Summary:**

[AI Review] This review covers BioTemporal-HAR, a proposal for a CNN-LSTM + multi-sensor fusion + SDT calibration pipeline for Human Activity Recognition (HAR) and biometric identification/verification on the WISDM dataset (51 subjects, 18 activities), scoped as a 4-week course project. Evaluated across three rounds at varying bars (NeurIPS, workshop, and course levels), the proposal is well-structured and leverages a principled SDT framing, but suffers from near-zero novelty, invalid citations, ungrounded targets, a lack of SOTA baselines, and an overly ambitious scope for the given timeline. A pivot focusing solely on SDT calibration for wearable biometric verification is recommended.

**Strengths:**

1. Well-scoped three-task formulation (activity recognition, identification, and verification).
2. Honest and transparent limitations section.
3. Strong awareness of data leakage issues in the evaluation plan.
4. Principled and underused Signal Detection Theory (SDT) framing applied to HAR.
5. As a course project, the proposal is well-structured, clear, and demonstrates a solid research plan.

**Weaknesses:**

1. Near-zero novelty: Every component relies on off-the-shelf methods (e.g., CNN-LSTM, standard multi-sensor attention, textbook feature selection) with no new theoretical, algorithmic, or architectural innovations.
2. Invalid citation: The 'Walrus Optimizer' reference lacks authors, a venue, a DOI, or peer review, which is a critical flaw in any academic context.
3. No SOTA comparison: The evaluation relies on straw-man baselines rather than comparing against published WISDM benchmarks, Transformers, Temporal ConvNets, or self-supervised methods.
4. Ungrounded research targets: Projected accuracies (e.g., 95% activity, 90% identification) are aspirational rather than hypothesis-grounded, lacking citation of prior results to justify these numbers.
5. Overstated SDT contribution: Applying d′ to verification scores is standard in biometrics, and the proposal does not address the equal-variance Gaussian assumption or articulate what SDT adds beyond standard DET curves/EER analysis.
6. Unrealistic timeline: Attempting to cover 3 tasks with multiple models, feature selection, SDT, and ablations in 4 weeks carries a high risk of producing superficial results.
7. Unexplored 20 Hz limitation: The low sampling rate fundamentally limits biometric discrimination but is not discussed as a constraint.
8. Activity-identity confound: The core technical challenge of how motion variance across activities affects identity is barely discussed.
9. Missing related work: Only 5 references are provided, with massive gaps in covering gait-based authentication, continuous authentication, and multi-sensor HAR surveys.
10. Open-set vs. closed-set evaluation flaw: Drawing impostor claims from held-out windows of known subjects still constitutes a closed-set test, failing to represent true verification.

**Questions:**

1. How do you plan to address the complete lack of novelty relative to decade-old baselines like Ordonez & Roggen (2016)?
2. Can you provide a valid, peer-reviewed replacement for the 'Walrus Optimizer' citation?
3. How will you ground your target accuracy metrics in prior state-of-the-art literature on the WISDM dataset?
4. What is your strategy for handling the activity-identity confound in your multi-task pipeline?
5. How do you intend to mitigate the fundamental biometric limitations imposed by the 20 Hz sampling rate?
6. Can you design a true open-set verification test using completely unseen impostors, rather than held-out data from known subjects?
7. Would you consider narrowing the project scope exclusively to 'SDT-Calibrated Wearable Biometric Verification' to ensure deeper, more feasible results within the 4-week timeline?

---

### Official Review · Reviewer_Bnft · 2026-05-19

**Rating:** 8
**Confidence:** 5

**Summary:**

This project aims to create a lightweight, multi-sensor framework for both activity recognition and behavioral biometrics (identity verification) using smartphone and smartwatch data. The authors propose using a compact CNN-LSTM model combined with Signal Detection Theory (SDT) to improve accuracy and balance decision thresholds for practical, energy-efficient deployment on wearable devices.

**Strengths:**

Practical Focus: The project prioritizes "lightweight" models suitable for real-world, low-power wearable devices rather than just maximizing performance.
Principled Decision Making: Incorporating Signal Detection Theory is an excellent way to handle the trade-off between security (false alarms) and usability.
Dual-Purpose Approach: Treating inertial signals as both activity indicators and biometric evidence is a sophisticated and highly relevant research direction.
Robust Evaluation Plan: The plan to use "leakage-safe" splits (grouped by subject/time) shows a strong understanding of how to avoid overfitting in time-series data.

**Weaknesses:**

Dataset Constraints: As the authors rightly note, the WISDM dataset is controlled and may not reflect the messiness of real-world "in-the-wild" usage.
Complex Scope: Balancing three distinct tasks (activity recognition, closed-set identification, and verification) is ambitious for a single project. There is a risk that one of these tasks might receive insufficient attention.
Privacy Implications: The proposal acknowledges privacy concerns, but given the sensitive nature of continuous biometric monitoring, this requires a very clear plan for how these models will respect user privacy.

**Questions:**

How do you plan to handle the situation where an activity's motion signature is similar to an impostor's gait, making it difficult for the model to distinguish between the two?
Since you are targeting edge deployment, what specific metric will you use to define "lightweight" (e.g., latency, memory footprint, or total parameter count)?
If the "Walrus Optimizer" does not yield significant improvements over simpler methods, do you have a fallback strategy for feature selection?